

# Multi-model Impacts of Climate Change on Pollution Transport from Global Emission Source Regions

Ruth M. Doherty[1], Clara Orbe[2], Guang Zeng[3], Michael Prather[4], David A. Plummer[5], Meiyun Lin[6], Drew Shindell[7], Ian A. Mackenzie[1], Oliver Wild[8]

[1]School of GeoSciences, University of Edinburgh, UK
[2]Goddard Earth Sciences Technology and Research (GESTAR); Johns Hopkins University, USA
[3]National Institute of Water and Atmospheric Research, Wellington, New Zealand
[4]Department of Earth System Science, University of California, Irvine, CA 92697-3100, USA
[5]Canadian Centre for Climate Modelling and Analysis, Environment and Climate Change Canada, Montréal, QC, Canada
[6]Program in Atmospheric and Oceanic Sciences at Princeton University and NOAA Geophysical Fluid Dynamics Laboratory, Princeton, NJ 08540, USA
[7]Nicholas School of the Environment, Duke University, Durham, NC 27708, USA
[8]Lancaster Environment Centre, Lancaster University, UK

*Correspondence to*: Ruth M Doherty (ruth.doherty@ed.ac.uk)

**Abstract.** The impacts of climate change on tropospheric transport, diagnosed from a carbon monoxide (CO)-like tracer species emitted from global CO sources, are evaluated from an ensemble of four chemistry-climate model (CCMs) contributing to the Atmospheric Chemistry and Climate Model Intercomparison Project (ACCMIP). Model time-slice simulations for present-day and end of the 21st century conditions were performed under the Representative Concentrations Pathways (RCP) climate scenario RCP 8.5. All simulations reveal a strong seasonality in transport, especially over the tropics. The highest CO-tracer mixing ratios aloft occur during boreal winter when strong vertical transport is co-located with biomass burning emission source regions. A consistent and robust decrease in future CO-tracer mixing ratios throughout most of the troposphere, especially in the tropics, and an increase around the tropopause is found across the four CCMs in both winter and summer. Decreases in CO-tracer mixing ratios in the tropical troposphere are associated with reduced convective mass fluxes in this region, which in turn may reflect a weaker Hadley Cell circulation in the future climate. Increases in CO-tracer mixing ratios near the tropopause are largely attributable to a rise in tropopause height, although a poleward shift in the midlatitude jets may also play an important role in the extra-tropical upper troposphere. An increase in CO-tracer mixing ratios also occurs near the Equator, centred over Equatorial/Central Africa, extending from the surface to the mid troposphere which is most likely related to localised decreases in convection in the vicinity of the Intertropical Convergence Zone, resulting in larger CO-tracer mixing ratios over biomass burning regions and smaller mixing ratios downwind.


# 1 Introduction

The transport of pollutants from the atmospheric boundary layer is governed by meteorological processes including deep convection, Hadley Cell driven overturning in the tropics, mid-latitude cyclones as well as slow low-altitude airflow, small-scale turbulent mixing and other motions (e.g., Cooper et al. 2011; TF-HTAP 2011). Climate change may affect the large-scale circulation of the atmosphere through the above processes, and hence impact the intercontinental transport of pollutants. In addition to influencing meteorological transport processes, changes in climate will also modify the atmospheric chemical environment and pollutant lifetimes. To understand how these changes will influence future pollutant distributions, it is therefore important to disentangle the relative impacts of changes in chemistry, emissions and transport. The focus of this study is to quantify climate change impacts on atmospheric transport.

In the tropics, the Hadley Circulation determines the location of the intertropical convergence zone (ITCZ) (e.g. Kang et al. 2014). Deep convection and mean upwelling associated with the Hadley Circulation control transport processes influencing pollutant distributions. For example, satellite measurements from MOPITT reveal that deep convection during the Asian summer monsoon carried pollutants emitted at the surface aloft into the upper troposphere (Kar et al., 2004). Deep convective transport of biomass burning emissions into the middle and upper troposphere was observed over Brazil, during the TRACE A atmospheric chemistry field campaign experiment (Pickering et al., 1996).

Over the mid-latitudes, ascent of pollution from the surface to the mid-to-upper troposphere occurs along warm conveyor belt (WCB) airstreams embedded within synoptic-scale mid-latitude cyclones (Cooper et al. 2002, Brown-Steiner and Hess, 2011; Lin et al. 2012). Also, descent from the lower stratosphere and upper troposphere to the mid-troposphere can occur in the dry intrusion airstreams of cyclones (e.g. Langford et al., 2014; Knowland et al. 2015). This is also the main mechanism for stratosphere–troposphere exchange that occurs in the mid-latitudes, and which may extend to the surface in regions prone to deep stratospheric ozone intrusions (Lin et al., 2015). Descent to the surface from the upper or mid-troposphere primarily occurs through subsidence in anticyclones at the ends of the mid-latitude storm tracks or in the vicinity of the descending branch of the Hadley Cell (Stohl et al. 2002; Cooper et al. 2004). Deep convection is also important for lofting surface pollution in mid-latitude regions in summer when the landmass is warm. However, weaker descent and enhanced photochemical destruction in summer compared to other seasons may lessen the importance of deep convection in reducing surface pollution levels (Brown-Steiner and Hess, 2011).

Changes in climate may impact many of these tropical and mid-latitude transport processes, but the impact of these future changes on chemical composition remains unclear. In the tropics and subtropics a number of studies have shown a poleward expansion and weakening of the Hadley Cell circulation in response to future increases in greenhouse gases, which is most robust in boreal winter (Vecchi and Soden 2007; Lu et al. 2007, Ma et al. 2012, Levine and Schneider 2011; Williamson et al. 2013; Hwan Seo et al. 2014; Kang et al. 2014). While these studies posit that the weakening of the Hadley Cell is related



to a weakening of the meridional temperature gradient between the tropics and sub-tropics, other studies have invoked thermodynamic constraints to suggest that convective mass fluxes throughout the tropics may decrease in response to increasing greenhouse gases (Held and Soden, 2006).

As temperatures increase in the troposphere but decrease in the stratosphere in response to enhanced $CO_2$ concentrations, there is a decrease in static stability close to the tropopause that leads to an increase in its height (Manabe and Wetherald, 1975; Santer et al. 2003; Lorenz and De Weaver 2007; Kang et al. 2014). A higher tropopause may also be associated with a poleward expansion or widening of the Hadley Cell (Lu et al. 2007), but the mechanisms underlying this change remain unclear.

Over the mid-latitudes, there is a general consensus that the storm tracks will shift poleward in response to future increases in greenhouse gases, at least in the zonal mean (Yin et al. 2005, Bengtsson et al. 2006, Barnes et al. 2013, Christensen et al. 2013; Shaw et al. 2016). This poleward shift in the mid-latitude storm tracks has been dynamically linked to the weakening of the Hadley circulation in the tropics (Shaw et al. 2016) and to the rise in tropopause height (Lorenz and De Weaver, 2007). However, the zonally asymmetric and seasonally varying response of mid-latitude storm tracks to forced climate change is much less robust (Simpson et al. 2014; Shaw et al. 2016), partly due to interannual variability (Deser et al. 2012; Shepherd 2014).

In terms of pollutant transport, this shift in the mid-latitude storm track position has been related to reduced mid-latitude cyclone frequency leading to increased summertime surface $O_3$ pollution episodes over the eastern USA and Europe (Mickley et al., 2004; Forkel and Knoche, 2006; Murazaki and Hess, 2006; Leibensperger et al., 2008; Wu et al., 2008), although other studies do not report such changes in frequency (Racherla and Adams 2008; Lang and Waugh, 2011). The shift in mid-latitude storm tracks has also been related to changes in regional climate phenomena in particular the North Atlantic Oscillation (Ulbrich et al. 2009; Christensen et al. 2013), blocking anticyclone frequency (Masato et al. 2013), and the Pacific Decadal Oscillation (Lin et al., 2014; Allen et al., 2014). Ozone transport from the lower stratosphere to the troposphere will also be influenced by future changes in stratosphere-troposphere exchange, which is expected to increase under greenhouse gas warming owing to a strengthening of the Brewer Dobson circulation in the stratosphere (Butchart and Scaife, 2001; Neu et al. 2014).

Few studies have explicitly isolated the effects of climate change on pollutant transport from its effects on chemical processes (e.g. through enhanced chemical reaction rates or changes in natural climate-sensitive emissions). Idealised tracers from surface sources were used by Holzer and Boer (2001) to show increases in interhemispheric exchange times, mixing times and mean transit times of ~10% between 2000 and 2100. This study also showed that a slightly higher tropopause was associated with reduced cross-tropopause tracer gradients and a lower tropospheric average tracer mixing ratio (Holzer and Boer, 2001). More recently, Orbe et al. (2015) used idealized tracers of air-mass origin to track how future increases in greenhouse gases modify transport patterns extending from the northern midlatitude boundary layer into the Arctic. Using idealised tracers they diagnosed enhanced poleward transport of the mid-latitude air arising from the poleward migration of the mid-latitude storm tracks, as outlined above (Orbe et al. 2015).



Using a carbon monoxide (CO)-like tracer, which has present-day fossil fuel emissions as its source and loss by reaction with present-day OH, Mickley et al. (2004) related 5-10% enhancements at the high percentile values of summer CO-tracer distributions in the United States to reduced cyclone passage across southern Canada under a future 2050 climate compared to present-day. Several other studies have used idealised CO-tracers emitted from continental sources to investigate climate

variability and change impacts on pollutant transport (Shindell et al. 2008, Doherty et al. 2013, Lin et al. 2014; Monks et. al. 2015). Using a regional CO-like tracer with surface emissions from Asia held constant, Lin et al. (2014) examined the mean influence of Eurasian pollution over the subtropical North Pacific, influenced by the position of the sub-tropical jet and its decadal variability. Under the SRES A2 climate forcing scenario for the 2090s compared to the 2000s, distinct dipole patterns in the changes in surface CO-tracer mixing ratios were interpreted as a response to future shifts in regional

circulations within four continental regions and their outflow locations (Doherty et al. 2013).

A detailed analysis by Fang et al. (2011) used a global CO-like tracer with a first-order 25 day lifetime and global anthropogenic CO emissions to investigate changes in transport under the SRES A1B scenario between 1981-2000 and 2081-2100 using the GFDL-AM3 chemistry-climate model (CCM). They found that CO-tracer mixing ratios increased at the surface and decreased in the tropical free troposphere due to reduced convective mass fluxes, and that reduced CO-tracer

mixing ratios in the southern hemisphere were most likely a response to a weaker Hadley circulation and reduced interhemispheric exchange (Fang et al. 2011). A large increase in CO-tracer mixing ratios near the tropopause was suggested to arise from the upward migration of the tropopause (Fang et al. 2011). This study focussed on annual-mean distributions.

The aim of this paper is to explore the robustness of the changes in transport described above using one model, across an ensemble of CCMs participating in the recent Atmospheric Chemistry and Climate Model Intercomparison Project

(ACCMIP) using a globally emitted CO-tracer (Lamarque et al., 2013), and to gain new insights into seasonal variability and dynamical attribution. Section 2 describes the models used while section 3 discusses future changes in CO-tracer mixing ratios (with emissions held constant). Section 4 outlines the transport processes and circulation changes that most likely drive CO-tracer redistribution under climate change. Discussion and conclusions are presented in section 5.

## 2 Data Sets and Methods

In the ACCMIP model intercomparison, four global CCMs included a CO-like tracer emitted from global sources: UM-CAM, GISS-E2-R, CMAM, and STOC-HadAM3. A description of these models, including their chemistry, transport and configuration, can be found in Lamarque et al. (2013) and Young et al. (2013). The horizontal resolution of the models varied between 1.875° by 2.5° and 5° by 5°. Two of the models, UM-CAM and STOC-HadAM3, have the same driving GCM; however, their advection schemes differ substantially since STOC-HadAM3 is the only model to use a Lagrangian

approach to simulate transport processes. The simulations were performed using decadal-average monthly sea surface temperature and sea ice concentration distributions for two 10-year periods: present-day ("acchist" simulations) as represented by a period centred on the year 2000 (1996-2005), and a future projection under the latest IPCC Representative





Concentration Pathway (RCP) RCP8.5 scenario for 2090-2099. Note that each modelling group derived their own set of sea surface temperature and sea ice fields, typically from a closely-related coupled-ocean GCM. Under RCP 8.5 the increase in global mean surface temperature between 2081–2100, relative to 1986–2005 is projected to be 2.6-4.8°C averaged across all (~39) participating GCMs, (Collins et al. 2013). For the four models used here the global mean surface temperature change between 1996-2005 and 2090-2099 is 3.1°-4.6°C.

The CO-like tracer was implemented as a chemically inert species with monthly-varying emissions representing all global anthropogenic and biomass burning CO sources with a first-order decay lifetime of 50 days (Shindell et al., 2008; Fang et al., 2011; Doherty et al. 2013). This idealised tracer is relatively long-lived such that it can undergo interhemispheric transport and be used to diagnose how changes in transport from source regions affect the distributions of trace gas species with similar lifetimes (such as CO and $O_3$). Monthly CO-tracer fields were generated for two 10-year periods (1996-2005 and 2090-2099), and the four models used the same emissions data for 2001 for both time periods. Thus, for each CCM the differences in CO-tracer mixing ratio distributions between these two periods are due solely to how climate change affects transport from global emission regions. To establish whether the CO-tracer distributions in the present-day (1996-2005) and future (2090-2099) periods are significantly different, a Student t-test was performed using the 10 years of annual data for each period for each model grid cell; a p-value < 0.05 was used to determine statistical significance.

The Task Force for Hemispheric Transport of Air Pollution (TF-HTAP) CO-tracer emissions dataset was used in ACCMIP, which consists of annual mean anthropogenic emissions for the year 2001 from the RETRO project (Schultz and Rast 2007; http://www.retro.enes.org) and seasonally varying biomass burning emissions (injected into the models at, or near, the surface) from GFED version 2 (van der Werf et al. 2006; http://www.geo.vu.nl/~gwerf/GFED.htm). The major source regions for anthropogenic CO emissions are in the northern mid-latitudes with peak levels in East and South Asia (Fig 1). Unlike the anthropogenic emissions, the biomass burning emissions feature a strong seasonality, with high values over Equatorial Africa during December-January-February (DJF), peak values southward of the Equator in South America and Central Africa in June-July-August (JJA), and a stronger peak value in Southeast Asia during JJA (Fig 1). Biomass emissions during March-April-May (MAM) and September-October-November (SON) are significantly weaker (Fig. 1).

Temperature data from the four CCMs were used to calculate the thermal tropopause following the World Meteorological Organization (WMO) lapse rate definition as implemented by Reichler et al (2003) for gridded reanalysis data. The tropopause is defined as the lowest model level at which the lapse-rate decreases to 2°C/km, provided that the average lapse-rate between this level and higher levels does not exceed this. Convective mass flux and zonal (u) wind data (available for three of the four CCMs) was also used for qualitative attribution purposes.

## 3 CO tracer Redistributions in Response to Climate Change

The distribution of the CO-tracer in the troposphere under present-day conditions and its redistribution under the RCP8.5 climate scenario are now discussed. Similarities and differences across the four CCMs and with season are highlighted. Note



that the monthly-average atmospheric burden of the CO-tracer is identical for the 2000s and 2090s, as expected given the specified emissions and lifetime, so that the differences in mixing ratio discussed here result purely from a re-distribution due to changes in transport.

### 3.1 Present-day Distributions

For the present-day period (1995-2006), the CO-tracer distributions show the effect of deep convection in the tropics and synoptic and convective lifting over mid-latitudes. There is a strong seasonality in the CO-tracer distributions, which is driven by both the seasonality of CO source emissions (Fig. 1) and seasonal changes in transport. With the exception of middle and high latitudes the largest CO-tracer mixing ratios occur during boreal winter (DJF), hereinafter winter, (Fig. 2) compared to boreal summer (JJA), hereinafter summer (Fig. 3).

During winter, large CO-tracer mixing ratios are found near the Equator, with decadal-average values of more than 60 ppb extending from the surface to ~700 hPa, and up to 40 ppb in the mid-upper tropical troposphere in all four CCMs (Fig. 2). In summer CO-tracer mixing ratios are lower in the tropics and northern extra tropics extending to ~40°N (~40-50 ppb at the surface and 30-40 ppb in the mid- upper troposphere in the northern hemisphere; Fig. 3)) than in winter (Fig 2.). In contrast, in the northern middle and high latitudes CO-tracer mixing ratios are higher in summer (up to 50 ppb near the surface; Fig

3.) than in winter (Fig 2.). Note that, while the CO-tracer distribution patterns are fairly similar between the models, there are some differences. In particular, CMAM simulates slightly lower values in the tropical upper troposphere in winter compared to the other CCMs (Fig. 2c), while the GISS-E2-R simulation features larger values above 700hPa over northern mid-latitudes during summer (Fig. 3d).

The spatial pattern of the CO-tracer averaged over the lower-mid troposphere (400-800 hPa; Fig. 4) is zonally relatively

uniform in winter and is similar across the four CCMs. This pattern also highlights the influence of strong vertical transport in the tropics and subsequent horizontal transport of the CO-tracer from its major anthropogenic surface sources over East and South Asia and biomass burning sources in Equatorial and Central Africa. During summer, the lower-mid tropospheric CO-tracer patterns are more closely confined to the source region locations over East/South Asia and Central Africa (Fig. 5) suggesting weaker transport around the middle troposphere than in winter.

The seasonal differences in CO-tracer mixing ratios in the tropics reflect the combined influence of seasonal differences in biomass burning emissions and in tropical convection. In particular, during winter, when the Intertropical Convergence Zone (ITCZ) is located in the Southern Hemisphere, the largest emissions from biomass burning originate over Africa near the equator (Fig. 1). In contrast, during summer, when the ITCZ is located north of the equator, the peak biomass burning emissions are located further southward (~20°S) (Fig. 1). Thus, in summer tropical convection is weakly collocated with

biomass burning emissions, resulting in lower CO-tracer mixing ratios in the tropical mid-troposphere as a whole and a more confined region of peak CO-tracer mixing ratios over Central Africa (Fig. 5), compared to winter (Fig. 4). The influence of the ITCZ position is also seen in the CO-tracer vertical gradient (the ratio of the CO-tracer relative to its value at the surface) which is shallower in the southern tropics during winter and in the northern tropics in summer (not shown) due to greater





vertical mixing from convective lofting along the ITCZ (section 4.1). Note that seasonal differences related to the transport of anthropogenic sources in the mid-latitudes may also be important, with strong surface lofting in winter storm tracks resulting in larger CO-tracer mixing ratios over the northern extratropics during winter (Fig. 4) than in summer (Fig. 5). The concentrations of CO-tracer are significantly lower during boreal spring (MAM) and autumn (SON) (not shown).

Henceforth, therefore, the focus is on findings for boreal winter and summer.

Note that the CO-tracer mixing ratios in this study are higher than those reported by Fang et al. (2011) since a longer lifetime of 50 days compared to 25 days is used in our study. CO-tracer mixing ratios are also typically lower than modelled or observed real CO (usually more than 100 ppb over source regions) as there is no chemical production of CO-tracer, which accounts for about half of atmospheric CO (Shindell et al. 2006).

**3.2 Response to Greenhouse Gas Increases**

The response of CO-tracer mixing ratios to climate change shows robust features that are statistically significant across the four CCMs (Figures 2 and 3). This suggests a general consistency in changes in transport between 1995-2006 and 2090-2099 under the RCP 8.5 climate scenario. In general, in both winter and summer, CO-tracer mixing ratios decrease through most of the troposphere. The largest changes occur in boreal winter with decreases of ~2-6 ppb (~5-10%) in CO-tracer mixing

ratios near the surface at the Equator and especially in the middle to upper tropical troposphere in the tropics and the northern mid-latitudes (Fig. 2). In contrast, there is a narrow region of increases in CO-tracer mixing ratios of up to 6 ppb (~10%) at ~5-10° N reaching from the surface to the mid-troposphere (and into the upper troposphere in CMAM) (Fig. 2). This feature is also seen in the annual-mean CO-tracer distributions for the four CCMs (not shown). Fang et al. (2011) find substantial but more widespread increases in annual-mean CO-tracer mixing ratios near the surface that extend from the

Equator to the northern mid-latitudes. Since the same emission data are used in both studies this difference is likely to arise from model variability in representing shallow convection and/or advection processes. Future CO-tracer mixing ratios also increase substantially by ~2-6 ppb (~10-25%) near the tropopause and into the lower stratosphere (where the relative changes can reach 50%), especially in the tropics and northern mid-latitudes in all the CCMs (Fig. 2). Similar difference patterns occur in summer, except that the narrow region of increase is above the surface and is smaller in magnitude and

vertical extent and is centred south of the Equator, where present-day CO-tracer concentrations peak (coinciding with the summer biomass burning peak) (Fig. 3). The fractional or relative changes in CO-tracer concentrations between winter and summer are fairly similar (not shown).

Examining the spatial changes in tropical CO-tracer concentrations in the lower to mid troposphere in relation to the increases and decreases described above, a clear dipole pattern emerges across all four CCMs (Figures 4 and 5). In

particular, during winter, a large increase centred over Equatorial and Central Africa (the regions with peak biomass burning) and a decrease south of this of similar magnitude of ~15 ppb (and up to 30 ppb for the region of decrease in UM-CAM and STOC-HadAM3) (Fig. 4) are shown. The area of increase coincides with the increase near the Equator extending from the surface to the mid-troposphere, seen in Figures 2 and 3 described above. This dipole pattern reflects stronger



confinement of CO-tracer mixing ratios near regions of emissions (Figures 4 and 5), flanked by smaller concentrations downwind to the south. Note that, while the zonally varying pattern of the response in the CO-tracer is characterised by high internal variability, this dipole pattern is statistically significant across most of its extent in all of the CCMs (Figures 4, 5). However, there are small differences across models. Similar patterns of change are simulated by the UM-CAM and STOC-

HadAM3 CCMs (Figures 4e-f, 5e-f) since they use the same driving GCM. A stronger and more extensive area of increase and a weaker area of decrease is simulated by the CMAM model in both seasons (Figures 4c, 5c), while GISS-E2-R simulates a weaker area of increase and a more extensive area of decreases that extends latitudinally across to S. America in winter (Figures 4d). Both CMAM and GISS-E2-R also depict an area of decrease over East Asia, although these changes are not statistically significant.

**4.  Potential Drivers of Changes in Transport in a Future Climate**

The impact of climate change under the RCP 8.5 scenario on deep convection and on jet-stream locations are outlined here in relation to the seasonal CO-tracer redistributions described above. The increase in tropopause height under $CO_2$ warming and its influence on CO-tracer distributions is also elucidated. Modifications to these transport processes have implications for pollutant transport from major source regions in the future.

**4.1  Convection and Jet-streams**

The meteorological and physical drivers of the CO tracer mixing ratios and changes due to climate change are examined using available data from the ACCMIP simulations. During both winter and summer, deep convection in the tropics extends from the surface to ~300hPa or higher in all four CCMs (Figures 6 and 7). While the magnitude of the (parameterized) convective mass fluxes simulated by the CMAM and especially the GISS-E2-R CCM (up to $30\times10^{-3}$ kgm$^{-2}$s$^{-1}$) are larger than

for UM-CAM and STOC-HadAM3 which are driven by the same GCM (up to $8\times10^{-3}$ kgm$^{-2}$s$^{-1}$), the spatial patterns and seasonalities of convection are consistent across all models (Figures 6 and 7). For example, in the tropics and subtropics, the strongest convective mass fluxes shift location from 10°-20°S in winter to 10°-20°N in summer as the Intertropical convergence Zone (ITCZ) migrates south and northwards of the Equator, as described in section 3.1. Convective mass fluxes are also large in the northern mid-latitudes in winter and in the southern mid-latitudes in summer when the mid-latitude jet-

streams are strongest, but their vertical extent is shallower than in the tropics.

The spatial patterns of convective mass fluxes averaged over the lower to upper troposphere (800 to 300 hPa) highlight the zonal symmetry of deep convection across the ITCZ, and depict the seasonal migration of the ITCZ within the sub-tropics (Figures 8 and 9). Hence, as discussed in section 3.1, the large values of CO-tracer mixing ratios over Equatorial and Central Africa during winter (Fig. 4) reflect the strong co-location of biomass burning emissions and convection in the Southern

Hemisphere subtropics when the ITCZ has shifted south of the equator. By comparison, lower CO-tracer mixing ratios over Equatorial Africa during summer reflect both a southward migration in emissions and a northward migration of the ITCZ,



resulting in weaker convective lofting in this region (Fig. 5). The larger CO-tracer mixing ratios at higher altitudes over northern mid-latitudes during summer simulated by the GISS-E2-R CCM (Figure 2d) may be related to stronger convective mass fluxes at this altitude (Fig. 7d).

A robust feature across all of the models is an overall reduction in convection in response to climate change throughout most of the troposphere in both winter and summer (up to $3\text{-}5\times10^{-3}$ kgm$^{-2}$s$^{-1}$; ~10-30%) that is slightly larger for UM-CAM and STOC-HadAM3 than the other two CCMs (Figs. 6 and 7). Absolute and relative changes in convective mass fluxes between winter and summer are similar (not shown). This reduction occurs both in the tropics and in the extra-tropics, extending to about 40°N/S. The UM-CAM and STOC-HadAM3 CCMs (driven by the same GCM) also feature strong decreases in convection centred at 60°N and 60°S extending vertically from the surface to 700 hPa in both seasons. The spatial pattern of the magnitude of convective mass fluxes averaged over the mid-upper troposphere shows the largest decreases occurring along the ITCZ (including the ITCZ portion over Africa) in both seasons for all CCMs (Figs 8 and 9).

Although convective mass fluxes predominately decrease under greenhouse-gas warming, there are small areas of increase. Close to the surface in the tropics increases in convective mass fluxes are simulated by the UM-CAM and STOC-HadAM3 CCMs, but not by the other two CCMs (Figure 6 a-b, 7a-b). The GISS-E2-R CCM depicts a small band of increased convective mass fluxes at the Equator extending from the surface to the upper troposphere in winter (Figure 6d). A strong increase in convective mass fluxes in the northern polar latitudes at the surface extending upwards is also a consistent model feature in winter, which is most prominent in UM-CAM and STOC-HadAM3 (Figure 6a-b). Since this region is not-collocated with major emissions, these increases in convection have a very small influence on the CO-tracer distributions. Increases in convective mass fluxes are also simulated over the East Pacific portion of the ITCZ by the GISS-E2-R and, to a lesser extent, by the CMAM models (Fig. 8 c-d, 9c-d). Again, since this area of increase is over the ocean and hence not co-located with emissions, there is little influence on the spatial patterns of CO tracer concentrations change (Figure 4 c-d, 5c-d).

The decrease in convective mass fluxes in the tropics under climate change described above is consistent with the reduced convective lofting of biomass burning emissions and, hence, decreased CO-tracer mixing ratios in the tropical mid to upper troposphere in all of the CCMs (Figures 2 and 3; section 3.2). Furthermore, reduced convection in the tropics may also explain the dipole pattern of change in CO-tracer mixing ratios over Equatorial/Central Africa in the lower to mid-troposphere, through greater confinement of CO-tracer concentrations to the region directly aloft of the surface emissions source (Figure 4 and 5; section 3.2). These changes in CO–tracer concentrations are primarily determined by the extent of co-location between convective and biomass burning source regions (section 3.1), and how convection changes over these emission source regions.

Reduced convective mass fluxes in the future may also partly explain decreases in CO-tracer concentrations in the mid-latitude mid to upper troposphere (Figure 2 and 3), although changes in the mid-latitude storm tracks may also play an important role in modulating the CO-tracer changes at these higher altitudes. In particular, all of the models feature a poleward shift in their zonal-mean zonal winds under climate change, as found in previous studies (e.g. Yin et al. 2005; Orbe



et al. 2015), leading to a reduction in zonal-mean winds in adjacent regions of the mid-latitude troposphere, in both seasons although generally largest in winter (Fig. 10). However, there are different responses for the different models with UM-CAM and CMAM featuring substantially weaker (~5 ms⁻¹; Fig 10a-b) zonal-mean winds poleward of ~35°S in winter, while the GISS-E2-R CCM shows weaker (5 ms⁻¹; Fig. 10c) zonal-mean winds poleward of at ~35°N in winter and ~35°N in summer.

In the tropics, the zonal-mean wind response to climate change are rather variable.  Zonally varying zonal wind fields were not archived in the ACCMIP simulations, which prohibits further investigation into the relationship between mid-latitude jet stream changes and CO-tracer responses over the mid-latitudes.

### 4.2 Tropopause Height

Robust increases in CO-tracer mixing ratios near the tropopause in the tropical and northern mid-latitudes in response to
climate change are seen in all four CCMs (Figs 2 and 3). Previous studies have attributed similar CO-tracer changes to changes in tropopause height (Fang et al. (2011) which is a robust feature of greenhouse-gas warming (e.g., Kang et al. 2014), and shown by all four CCMs (Figs 2 and 3). Following Fang et al. (2011), we elucidate the role of an increase in tropopause height in modulating CO-tracer concentrations by comparing annual-mean profiles of the CO-tracer, plotted relative to the thermal tropopause, for both present-day and future periods (Fig. 11).

Comparisons of annual-mean CO-tracer profiles, plotted both relative to pressure and tropopause height, reveal that much of the positive CO-tracer increase near the tropopause that occurs in the future simulations by the CCMs reflect a rise in tropopause height (Fig. 11). This holds for all of the models over both the Equator and the mid-latitudes (40°S and 40°N). Hence the increase in CO-tracer mixing ratios arises from a transition between low-CO stratospheric air for present-day and higher-CO in tropospheric air in the future. However, after accounting for the influence of a higher tropopause, tropopause-
relative profiles, in the RCP 8.5 climate simulations in the 2090s, have a weaker vertical gradient in the NH extra-tropics, compared to the present-day simulations in the 2000s (Fig. 11, top panels). This may reflect an overall increase in mixing related to changes in meridional eddies, which shift upward and poleward in concert with the zonal mean zonal winds (as described above) and which weaken the gradient of the CO-tracer between the surface and the upper troposphere (Wu et al. 2013; Orbe et al. 2015).

### 5. Discussion and Conclusions

In response to future increases in greenhouse gases in the 2090s under the RCP 8.5 scenario, changes in mixing ratios of a CO-like tracer with a 50-day lifetime exhibit robust features across four Chemistry-Climate Models (CCMs) participating in the ACCMIP model intercomparison: a) a decrease in CO-tracer mixing ratios throughout most of the troposphere, especially in the tropics, and b) an increase in CO-tracer mixing ratios near the tropopause, especially over the tropics and
northern mid-latitudes. Underlying these changes there is a strong seasonality in transport patterns between winter and



summer, with higher CO-tracer concentrations aloft in winter when biomass burning emissions source regions are located along the ITCZ.

These changes in CO-tracer concentrations due to climate change are generally larger in boreal winter than in summer, although relative changes are similar. These features have previously been reported by Fang et al. (2011) using the GFDL-AM3 CCM. In addition, at the surface, all four CCMs simulate a small region of increase in zonal-mean CO-tracer mixing ratios at ~5-10°N that extends to the mid troposphere, with decreases southwards, which arise from a dipole pattern of adjacent increases and decreases in CO-tracer mixing ratios over Equatorial and Central Africa – the largest biomass burning emission source regions.

Convective mass fluxes consistently decrease throughout most of the troposphere in the future in all four CCMs in both seasons, with the strongest decreases occurring within the tropics along the ITCZ, consistent with a weakening of the Hadley Cell- at least in winter when this feature is robust (Hwan-Seo et al. 2014; Kang et al. 2014). The decreases in CO-tracer mixing ratios in the tropical troposphere therefore most likely reflect reduced convection in the future. Reduced convection in the tropics may also explain the dipole in CO-tracer response that occurs near the Equatorial and Central Africa, since the seasonal patterns of changes in CO–tracer concentrations in the tropics are in essence determined by how seasonal changes in convection project onto seasonally-varying biomass burning emissions. Biomass burning emissions were held constant for present-day and future in this study.

The strong increases in CO-tracer concentrations in the vicinity of the tropopause can be largely attributed to a higher tropopause under climate change, in accord with Fang et al. (2011). Changes in mixing in the upper troposphere, associated with a poleward and upward shift in zonal-mean winds (a response that is consistent across the four models), may also be important in interpreting some of the near-tropopause changes in the CO-tracer. Overall, large-scale dynamical responses linked to changes in the Hadley Cell circulation and their impact on convection, mid-latitude jets and in tropopause height appear to govern the main features of the redistribution of CO-tracer mixing ratios between present-day and future simulated by four CCMs in this study. Further diagnostics to allow more detailed dynamical insights would be useful to probe the relative contributions of different large-scale dynamical processes, alongside other aspects of the Hadley Circulation such as its broadening or poleward expansion under climate change.

One further limitation of this study is that the CO-tracer data is not available at a higher temporal resolution than monthly. Hence, it is not possible to examine how CO-tracer concentrations in the mid-latitudes are influenced by changes in synoptic-scale storm or blocking frequency. Our ability to look further into relationships between the mid-latitude storm tracks and CO-tracer distributions over middle and high latitudes is also limited by the fact that only zonally average zonal wind fields were available as output. Another limitation is that, while the use of a single tracer emitted from all global CO-sources highlights transport associated with the global continental emission source regions, it precludes an in-depth analysis of regional changes in transport patterns under climate change. Future work performed for the Chemistry Climate Model Initiative (Eyring et al., 2013) using seasonally invariant emissions will aid to isolate the effects of seasonality in transport from seasonality in emissions, to enable further insights into climate changes impacts on transport.



Nevertheless, this multi-model study presents a clear and robust picture of the effect of climate change on the transport of pollution from major emission source regions, and how this effect varies seasonally as governed by the seasonal location of the ITCZ and biomass burning emissions sources. Furthermore, the key roles of reduced convection consistent with a weakened Hadley Circulation in winter and a higher tropopause in governing transport changes are confirmed. Overall, a

reduction in tropical deep convection under climate change will confine pollution more closely to its surface source regions, potentially reducing inter-continental transport in upper-level winds aloft. In the mid-latitudes transport of pollution aloft will be impacted by a poleward shift in storm track pathways. Understanding these dynamical changes will aid to characterise the response of global pollution transport to greenhouse gas warming.

*Data availability*. The ACCMIP data used in this study can be obtained from the British Atmospheric Data Centre (BADC) upon request.

*Competing interests*. There are no competing interests.

*Author contribution*. RD, CO, GZ, MP, DP, ML contributed to conception and design. GZ, DP, DS and IM performed ACCMIP simulations with the UM-CAM, CMAM, GISS-ER-2, and STOC-HadAM3 CCMs respectively. CO, RD, GZ and IM contributed to processing of data. All co-authors contributed to the analysis and interpretation of data. RD and CO drafted the article aided with revisions by GZ, DP, ML, IM, OW. All co-authors approved the submitted version for publication.

*Acknowledgements*.  Ruth Doherty and Ian MacKenzie acknowledge ARCHER, the UK national high-performance computing service which is funded by the Office of Science and Technology through EPSRC High End Computing Programme, and funding under the UK Natural Environment Research Council grant NE/I008063/1, NE/M003906. GZ acknowledges the contribution of NeSI high-performance computing facilities to the results of this research. NZ's national

facilities are provided by the NZ eScience Infrastructure and funded jointly by NeSI's collaborator institutions and through the Ministry of Business, Innovation & Employment's Research Infrastructure programme (https://www.nesi.org.nz). ACCMIP is organized under the auspices of Atmospheric Chemistry and Climate (AC&C), a project of International Global Atmospheric Chemistry (IGAC) and Stratospheric Processes And their Role in Climate (SPARC) under the International Geosphere-Biosphere Programme (IGBP) and World Climate Research Program

(WCRP). The authors are grateful to the British Atmospheric Data Centre (BADC), which is part of the NERC National Centre for Atmospheric Science (NCAS), for collecting and archiving the ACCMIP data, and to Jean Francois Lamarque for overall ACMIP co-ordination.



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



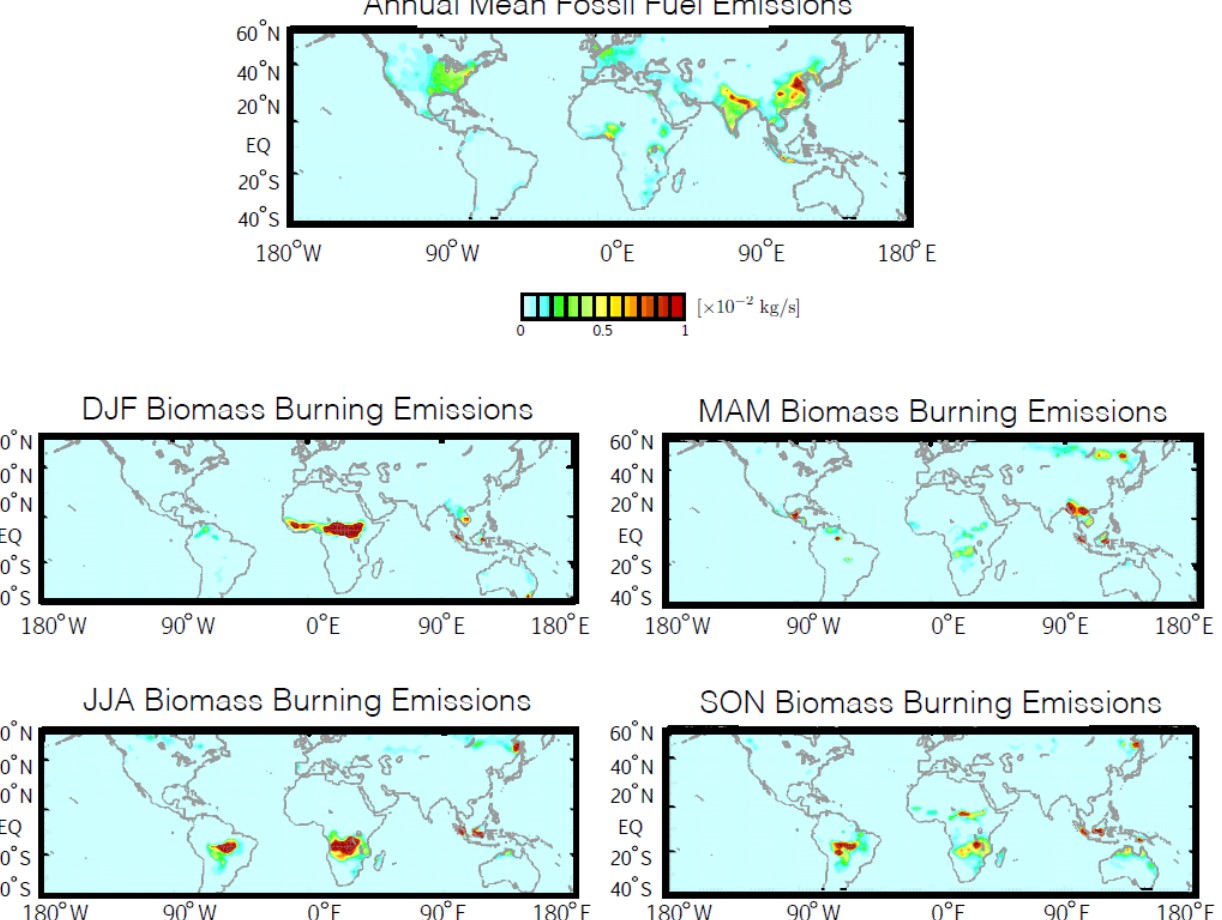

**Figure 1.** Annual mean anthropogenic CO emissions (kgs$^{-1}$) from fossil fuels (top panel) and seasonal biomass burning CO emissions (kgs$^{-1}$) (middle/bottom panels).





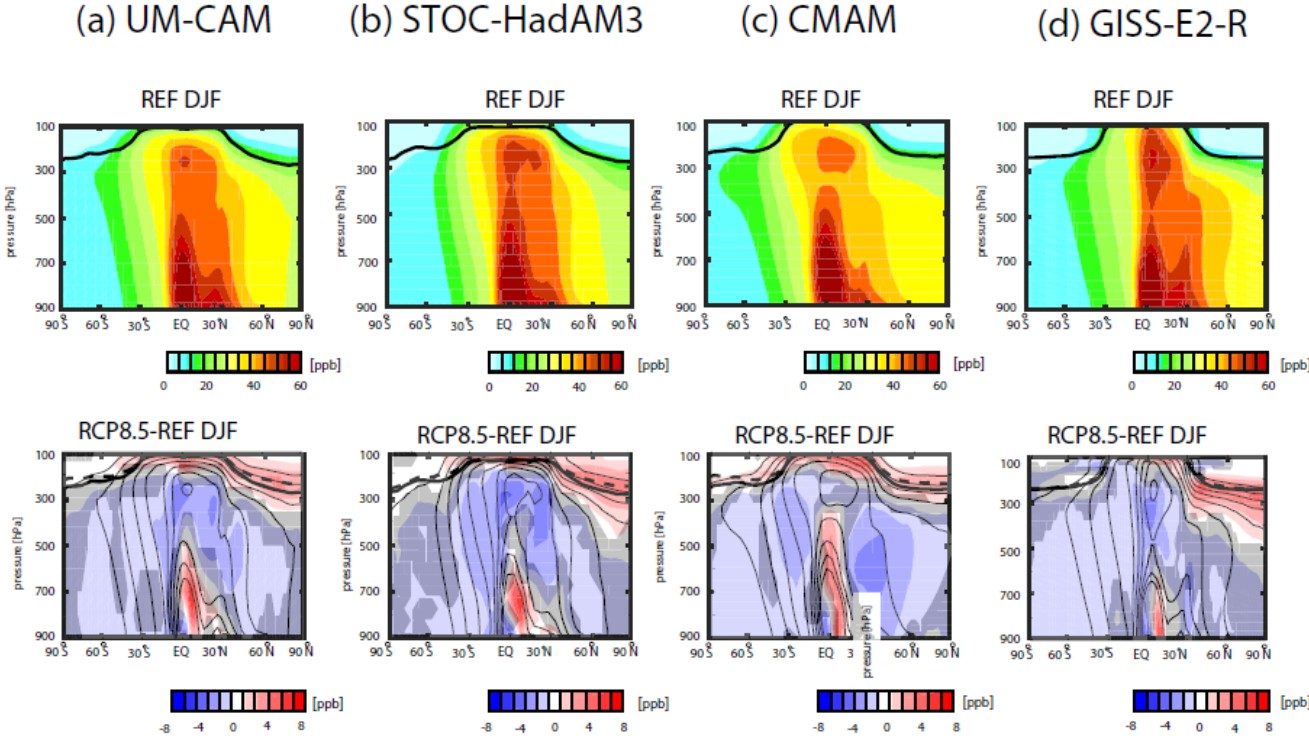

**Figure 2.** Top panels: Present-day (1995-2006) or REF boreal winter (DJF) climatological mean zonal mean CO distribution for a) UM-CAM, b) STOC-HadAM3, c) CMAM and d) GISS-E2-R CCMs. The thick black solid line represents the present-day (DJF) zonally averaged thermal tropopause. Bottom panels: 2090-2099 (RCP 8.5) - 1995-2006s (present-day) differences in the DJF 10-year climatological mean zonal mean CO distribution. Thin black contours denote the present-day DJF climatology. The thick solid and dashed
10 lines represent the DJF zonally averaged thermal tropopause for present-day and the 2090s (RCP8.5) climatologies. Grey shading indicate where results are not significant at p < 0.05 as evaluated with a Student t-test using 10 years of data for the 2090s (RCP 8.5) and present-day climate simulations.





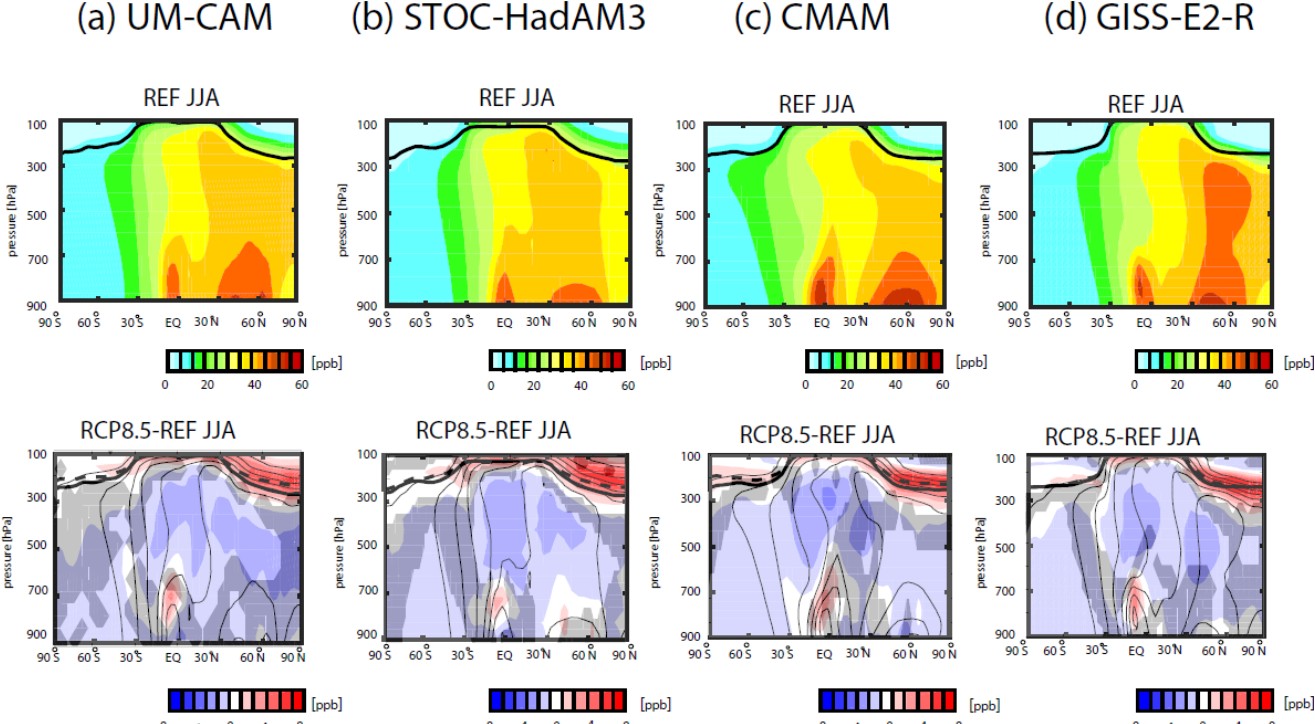

**Figure 3.** Same as Figure 2 but for Boreal summer (JJA).





**Figure 4.** Top panels within subplots a-d): Present-day (1995-2006) or REF DJF climatological mean CO distribution, averaged over 400-800 hPa. Bottom panels: 2090-2099 (RCP8.5) - 1995-2006 difference in DJF climatological mean distributions, wherein black contours denote the present-day climatology. Results are presented for a) UM-CAM and b) STOC-HadAM3 (top panels) and c) CMAM and d) GISS-E2-R (bottom panels). Grey shading indicate where results are not significant at $p < 0.05$ as evaluated with a Student t-test using 10 years of data for the 2090s (RCP 8.5) and present-day climate simulations. Note the different scales for UM-CAM and STOC-HadAM3 and for CMAM and GISS-E2-R for the difference plots.

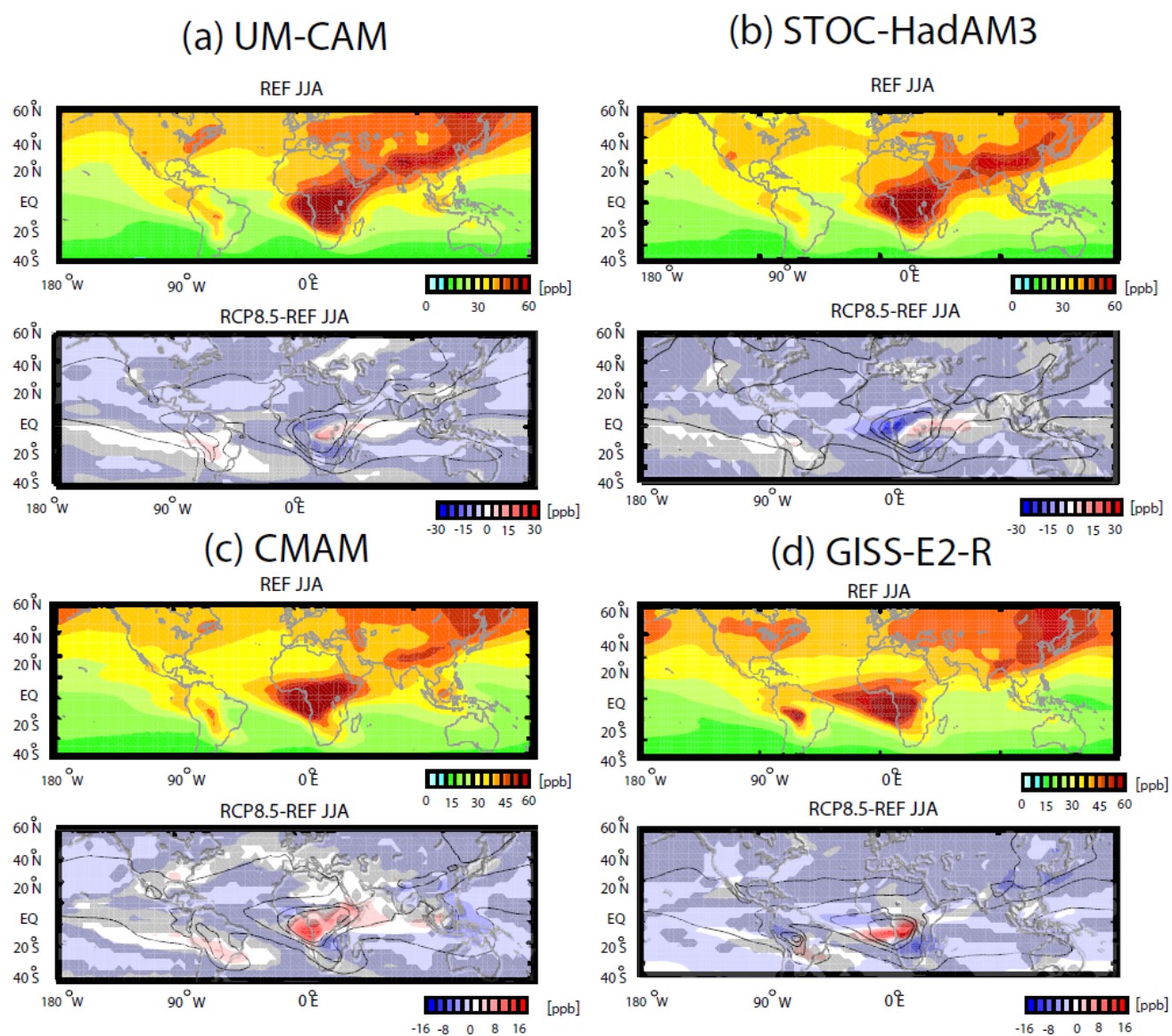

**Figure 5.** Same as Figure 4 but for JJA. Note the different scales for UM-CAM and STOC-HadAM3 and for CMAM and GISS-E2-R for the difference plots.





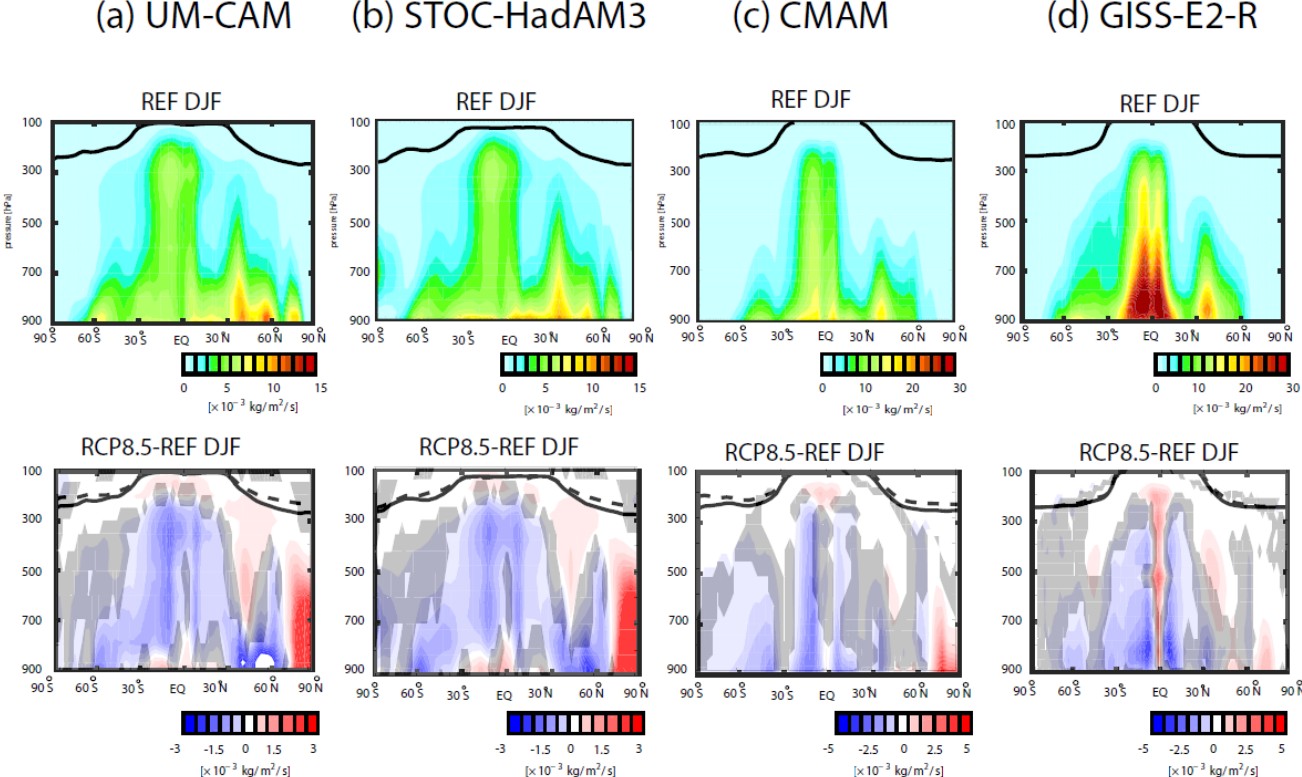

**Figure 6.** Top panels: DJF climatological mean zonally averaged convective mass fluxes for present-day (1995-2006) or REF for a) UM-CAM, b) STOC-HadAM3, c) CMAM and d) GISS-E2-R. The thick black solid line represents the present-day (DJF) zonally averaged thermal tropopause. Bottom panels: 2090-2099 (RCP8.5) -2000s differences in the DJF climatological mean zonal mean convective mass fluxes. The thick solid and dashed lines represent the DJF zonally averaged thermal tropopause for the present-day and the 2090s (RCP8.5) climatologies. Grey shading indicate where results are not significant at $p < 0.05$ as evaluated with a Student t-test using 10 years of data for the 2090s (RCP 8.5) and present-day climate simulations. Note the different scales for UM-CAM and STOC-HadAM3 and for CMAM and GISS-E2-R.





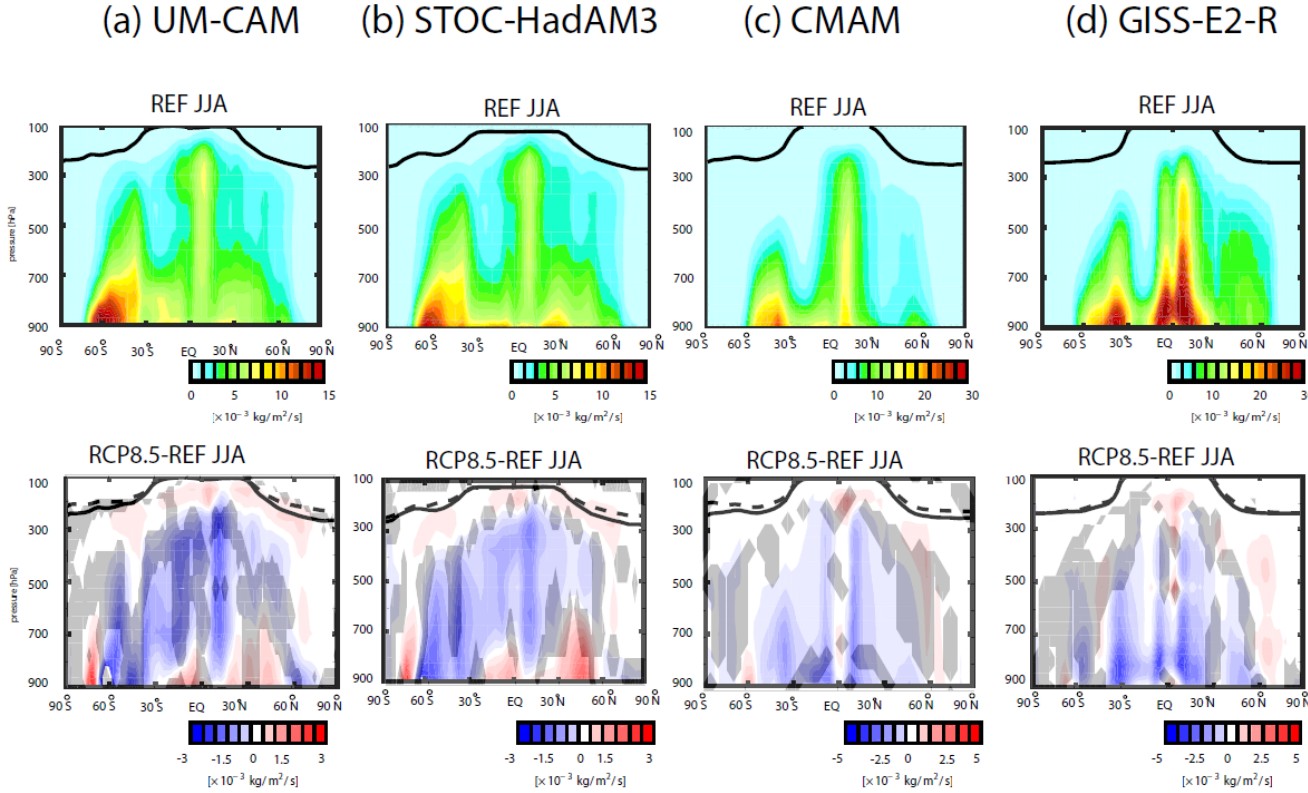

**Figure 7.** Same as Figure 6 but for JJA. Note the different scales for UM-CAM and STOC-HadAM3 and for CMAM and GISS-E2-R.







**Figure 8.** Top panels within subplots a)-d): DJF climatological mean convective mass fluxes, averaged over 300-800 hPa for 1996-2005 (present-day) or (REF). Bottom panels e)-h): Same, but for 2090-2099 (RCP8.5)- 1996-2005 (present-day) difference, wherein black contours denote the present-day climatology. Results are presented for UM-CAM and STOC-HadAM3 (top panels) and CMAM and GISS-E2-R (bottom panels). Grey shading indicate where results are not significant at $p < 0.05$ as evaluated with a Student t-test using 10 years of data for the 2090s (RCP 8.5) and present-day climate simulations.



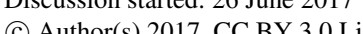


**Figure 9.** Same as Figure 8 but for JJA. Note the different scales for UM-CAM and STOC-HadAM3 and for CMAM and GISS-E2-R for the difference plots.





**Figure 10.** Zonal mean zonal wind 2090-2099 (RCP8.5) - 1996-2005 (present-day) or (REF) differences for DJF (top panels) and JJA (bottom panels). Results are shown for a) UM-CAM/STOC-HadAM3 (same driving GCM), b) CMAM and c) GISS-E2-R. Black contours denote the present-day climatology.





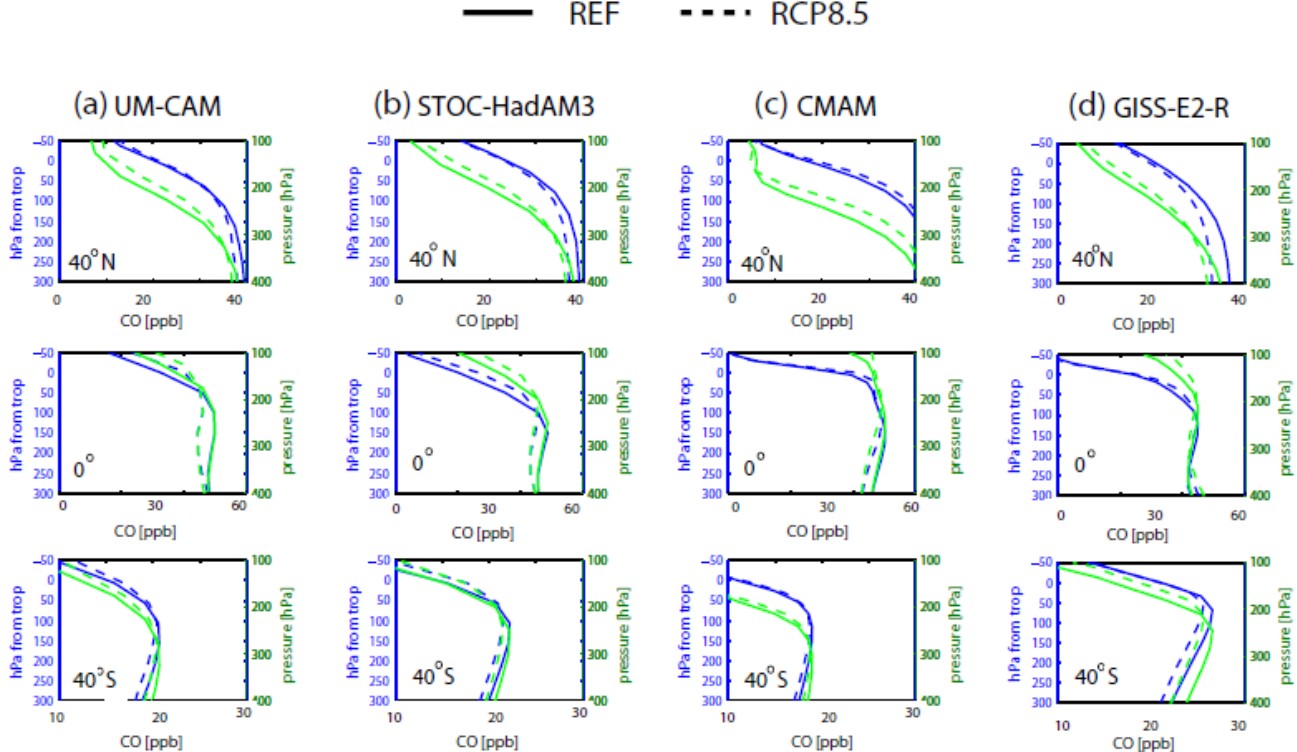

**Figure 11.** CO- tracer mixing ratio annual-average profiles (ppb) averaged over various latitude bands (40°N, 0, 40°S) for the 2000s (Present-day) or (REF) and the 2090-2099 (RCP8.5) for a) UM-CAM, b) STOC-HadAM3, c) CMAM and d) GISS-E2-R plotted against altitude (green) and with distance from the tropopause (blue).