# Peer review of "Multi-model Impacts of Climate Change on Pollution Transport from Global Emission Source Regions"

_Atmospheric Chemistry and Physics, 2017_

## Referee Comment (RC1) · Anonymous Referee #2 · 12 Jul 2017

The paper makes use of a global co tracer to track changes in advection patterns between present and future climate. The tracer has a fixed decay of 50 days. In this way the total CO mass is the same in both present and future climate, and any changes in in CO can be attributed to changes in advection.

This is an innovative use of tracers that can single out effects from pure advection processes as opposed to the combined effects of advection and chemistry etc in chemical tracer models. A major portion of the CO tracer is from biomass burning emitted in the tropics. Similar studies using purely anthropogenic tracers emitted at mid. latitudes are referred to in manuscript. In this way this study complements previous studies.

Major comments/suggestions:

I wonder if the zonally averaged figures (Figure 2-3, 6-7) would have been easier to interpret if the distance between the latitudes and the vertical axis had been scaled by mass?

Reading the paper the reader really has to keep the tongue straight in the mouth in order to follow all the effects of seasonal sources, convection etc. Maybe this is how it has to be, but could some sort of illustration/table help? Just a suggestion to be considered: Maybe a table or bar figure in section 2, "Data sets and methods" (page 4 - 5) with seasonal emissions split by region and natural vs anthropogenic emissions would help in the interpretation? The table/figure could in some way also be supplied with for example arrows of varying length, indicating the convective strength as this is a key feature in the interpretation of the data?

On page 12 the authors state that this study presents a clear and robust picture of the effect of climate change on the transport of pollution from major emission source regions. This is not quite true, as the major sources in this study are located in the tropics. This is partially true for CO, but not necessarily true for other air pollutants.

The authors should also state more clearly what knowledge has been gained, and in what way this study may improved out understanding of future air quality (and short lived climate forcers). Could there for instance be any potential feedback mechanisms to climate?

Minor corrections

Starting with the introduction (page 2) I see that for several citation the reference year in the text and the reference list don't match. Starting from page 2 (Introduction): TF-HTAP 2011 (2010), Kang et al. 2014 (2013), Cooper et al. 2002 (2004), Langford et al. 2014 (2015) Hwan Seo et al. 2014 - should it be Seo et al. 2014? Year in text and year in reference list in brackets. Please double check, and use the correct year both

places. NB! I have only checked the first page, and leave it to the authors to check the rest of the manuscript.

Page 3, line 10 Bengsson or Bengtssen (as in reference)?

Page 4, Confusing use of "one model" (line 18) and models in lines 19 - 23.
* * *

---

## Referee Comment (RC2) · Anonymous Referee #3 · 18 Jul 2017

Review of

Multi-model Impacts of Climate Change on Pollution Transport from Global Emission Source Regions

by Doherty et al.

Overview.

The authors compare present-day day air pollution transport patterns with projected end-of-21st-century conditions. For this purpose, seasonally-averaged volume mixing ratio fields of an artificial tracer simulated by four chemistry-climate-model of the

ACCMIP project are inter-compared. The tracer has a 50-day lifetime and is emitted using present-day anthropogenic and biomass burning CO emissions. The authors find a general decrease in tracer concentrations in the troposphere and an increase near the tropopause in the simulation of all models. The authors attribute this mainly to reduced convection in the tropics and an increase in tropopause height. The results agree across the four models and also with previous studies.

General comments

The work is a scientifically sound study of the impact of climate change. Decrease in convective activity, increase in tropopause height as well as weakening but increase in extend of the Hadley circulation is a climate response simulated by many CCMs. Showing the impact on concentration patterns as presented in the study is a step further in understanding the impact of these circulation changes.

The relevance and novelty aspect of the study could be enhanced by elaborating in more detail the agreement but also the disagreement with the studies from the literature, many of them mentioned in the introduction. Please include this in the conclusion section.

The presented study focuses a lot on the tropics. But changes in the transport to the Arctic is also an important aspect given the impact of black carbon on the radiative forcing. For example Orbe et al. (2015) and (2013) find enhanced pole-ward transport towards the Arctic. It would be good if the results of the current study would report more on the change in the poleward transport. Also, please show maps from 90S - 90N and do not omit the high latitudes.

To avoid misunderstandings, the authors should discuss more clearly the limitations of the study because of the use of the artificial tracers: i.e. no impact of the photochemistry (loss by OH and production by VOC) or the precipitation patterns (deposition), and , most importantly, no change in the emissions.

[Figure]

The end-of-century volume mixing ratios appear overall lower than the volume mixing ratios of the present day runs. But the total burdens should be the same ( p6 l1). It would be good if you could confirm the mass conservation and explain in more detail how the burden was redistributed. A comment on the mass conservation of the SL-advection scheme of STOC-HadAM3 might also be helpful.

The mechanism for the increase in tracer mixing ratios below the tropopause should be better explained. Is it simply because of the higher tropopause (i.e.the increase occurs near the present day tropopause) or do differences in stratosphere-troposphere exchange also play a role. Once it is established that the tropopause is higher from the GCM run, it is somewhat a trivial finding that tropospheric mixing ratio for primary tracers are increased at the same height.

Specific comments:

P 1 L 24 , I think the weaker Hadley cell is a result of the GCM calculations, i.e. a given for this study So please consider rephrasing " .. in turn reflect . . ." to " . . . causes . . . "

P1 L25 Please add a sentence on the mechanism of increase in tracer because of increase in tropopause height.

P2 L19-L27 This is more relevant for ozone and not so much for transport of primary pollutants. Consider shortening or omitting it.

P3 L 17 Is this response in ozone caused by transport ? If not please omit.

P3 L 23 Please discuss the impact of changes in stratosphere-troposphere exchange for the tracer transport

P3 L31 Please add also Orbe et al. (2013)

P4 L29 Please add also information about the different convection schemes of the 4 models

P5 L19 Please state the temporal resolution of the GFED 2 data

P7 L4 Please add a discussion here how the present-day CO tracers compare to actual CO. The NH CO maximum occurs in April, which seems not the case for tracer.

P7 L5 It is not clear that lower values do not deserve consideration. The changes could be even stronger. Please elaborate on this. MAM is the maximum of present day CO.

P7 L23 Please clarify if this is the thermal present-day tropopause or the tropopause for the respective time slice. How does the thermal tropopause relate to a "tracer" tropopause?

P8 L9 Please discuss also the changes at high latitudes. (see Orbe et al. 2013)

P8 L17 From all ACCMIP models or only the four discussed here ?

P8 L18 For the present-day or end-of-century runs ?

P8 L20 Please provide more explanation for the up to factor 4 differences in the convective fluxes by the models. Is it driven by the meteorological input (i.e. T profile) or the specifics of the parametrisation.

P8 L21 Is there any indication, which of the models simulates more realistic mass fluxes.

P9 L9 Please discuss also the increase in convection north of 60 N in DJF shown in Figure 6.

P10 L10 Please provide a plot of the increase in the thermal tropopause height between present-day and end-of-century or give some numbers in pressure and height.

P10 L15 Please discuss Fig 11 in more detail. Is the conclusion based on the fact that dotted and solid blue lines overlap more than the respective green lines? I am not sure if this is actually the case especially for the the tropics.

P10 L21 The weaker across-the-tropopause gradient is an interesting finding. It should be mentioned more clearly in the paper, i.e. the conclusions.

P11 L18 Please discuss also the response in the Arctic (compare with Orbe et al. (2013, 2015)) and the hemispheric gradient (Holzerand Boer, 2011)

P11 L22 Please mention that you found a weaker across-the-tropopause gradient (if this is the case)

Figures 4,5, 8 etc. please show maps from 90S-90N

Figure 1 add to caption "for different seasons"

Figure 2, please use either present-day or REF not both

Figure 11, please mention that the distance is from the respective tropopause for each time slice.

Literature: Orbe, C., M. Holzer, L. M. Polvani, and D. Waugh (2013), Air-mass origin as a diagnostic of tropospheric transport, J. Geophys. Res. Atmos. 118, 1459–1470, doi:10.1002/jgrd.50133

―――――――――――――――

---

## Author Comment (AC1) · 7 Oct 2017

We thank both reviewers for their thoughtful and insightful comments that have considerably improved the manuscript. We provide our responses to reviewer's comments below and describe changes made to the manuscript. Any other changes to the manuscript not noted here are purely editorial. The referees' comments are copied below in italics, with our responses in below and our modifications to the manuscript in quotes. In our revised manuscript, the modified text is shown using track changes, and the modified figures are highlighted.

Referee #2

*The paper makes use of a global co tracer to track changes in advection patterns between present and future climate. The tracer has a fixed decay of 50 days. In this way the total CO mass is the same in both present and future climate, and any changes in in CO can be attributed to changes in advection. This is an innovative use of tracers that can single out effects from pure advection processes as opposed to the combined effects of advection and chemistry etc in chemical tracer models. A major portion of the CO tracer is from biomass burning emitted in the tropics. Similar studies using purely anthropogenic tracers emitted at mid. latitudes are referred to in manuscript. In this way this study complements previous studies.*

We thank the reviewer for their positive comments.

*Major comments/suggestions:*

*I wonder if the zonally averaged figures (Figure 2-3, 6-7) would have been easier to interpret if the distance between the latitudes and the vertical axis had been scaled by mass?*

We have thought carefully about this comment. However we have not followed this recommendation for the following related reasons. 1) Scaling by mass would further highlight the tropics and the surface features of the CO-tracer in response to climate change there, which we feel are clear enough in our zonal-mean plots, but this scaling would capture less well the climate-change related features in the northern mid-latitudes near the tropopause which is an important result that we highlight. Moreover, we compare these prominent features of the CO-tracer response to climate change with results from previous studies such as Fang et al. (2011), that do not scale by mass. Hence we feel our comparison with previous studies would be less clear if we did scale by mass.

*Reading the paper the reader really has to keep the tongue straight in the mouth in order to follow all the effects of seasonal sources, convection etc. Maybe this is how it has to be, but could some sort of illustration/table help? Just a suggestion to be considered: Maybe a table or bar figure in section 2, "Data sets and methods" (page 4 - 5) with seasonal emissions split by region and natural vs anthropogenic emissions would help in the interpretation? The table/figure could in some way also be supplied with for example arrows of varying length, indicating the convective strength as this is a key feature in the interpretation of the data?*

We thank the reviewer for their suggestions. We feel the best way to highlight the connections between the seasonality in emissions and seasonality in convection is to add to Figure 1 a line to show the position of the ITCZ in the different seasons. We also explored the use of a multi-model mean contour of convective mass flux, but this was too noisy so we retained a line to display the ITCZ position. We have added the following text to section 2 "Data Sets and Methods" and to the caption of figure 1:

Page 5 line 34: "The location of these emission peaks in relation to the position of the Intertropical convergence zone (ITCZ) can be clearly seen (Fig 1). "

"Figure 1.....panels).The dashed line shows the approximate position of the Intertropical Convergence Zone (ITCZ) in the different seasons."

We also feel this addition to Figure 1 makes the existing text in section 3.1 (page 7 line 5) and thereafter, where we highlight connections between emissions and dynamical seasonal patterns and co-location clearer to understand.

*On page 12 the authors state that this study presents a clear and robust picture of the effect of climate change on the transport of pollution from major emission source regions.  This is not quite true, as the major sources in this study are located in the tropics. This is partially true for CO, but not necessarily true for other air pollutants.*

We agree our study highlights the major CO sources in the tropics and their interactions with deep convective lofting; however we do also highlight the ubiquitous decrease in CO-tracer in most of the troposphere due to climate change.  We have revised the text in the manuscript to highlight these points.

Page 7 line 29: "In general, in both winter and summer, CO-tracer mixing ratios decreases are ubiquitous throughout most of the troposphere."

Page 13 line 10: "Nevertheless, this multi-model study presents a clear and robust picture of the effect of climate change on the transport of pollution from major emission source regions, in particular from biomass burning regions in the tropics that are strong CO sources, and how this effect varies seasonally as governed by the seasonal location of the ITCZ and biomass burning emissions sources."

*The authors should also state more clearly what knowledge has been gained, and in what way this study may improved out understanding of future air quality (and short lived climate forcers). Could there for instance be any potential feedback mechanisms to climate?*

We are unable to discuss potential feedback mechanisms to climate as the chemistry-climate model simulations did not include this two-way interaction between chemistry and radiation. Without explicitly including this interaction in the model set-up we could not reliably discuss feedbacks to climate. We agree that the last sentence of the manuscript could be much clearer about knowledge gained, and our understanding of future air quality and we thank the reviewer for this suggestion. We have added additional text to the last paragraph of the discussion and conclusions:

Page 13, line 17: "Hence, considering the impact of transport changes alone in the absence of stricter emissions controls, air quality in the future has the potential to be worsened in the vicinity of emission source regions especially in the tropics, due to reductions in vertical transport and dispersion by deep convection. However, this study examines the impacts of climate change on transport alone. Future air quality will also be greatly influenced by climate-driven changes in chemistry and by future changes in emissions. Future multi-model comparison studies would benefit from a larger suite of meteorological variables that enable a more detailed diagnosis of the large-scale dynamical responses to climate change. Such improved dynamical attribution in tandem with tracer transport studies will permit a fuller quantification of the response of global air pollution transport to greenhouse gas warming."

*Minor corrections*

*Starting with the introduction (page 2) I see that for several citation the reference year in the text and the reference list don't match.  Starting from page 2 (Introduction): TF- HTAP 2011 (2010), Kang*

*et al. 2014 (2013), Cooper et al. 2002 (2004), Langford et al. 2014 (2015) Hwan Seo et al. 2014 -
should it be Seo et al. 2014? Year in text and year in reference list in brackets. Please double check,
and use the correct year both places. NB! I have only checked the first page, and leave it to the
authors to check the rest of the manuscript.*

We thank the reviewer for noting these citation errors. We have modified the text and checked
throughout.

*Page 3, line 10 Bengsson or Bengtssen (as in reference)?*

Amended to that in the reference.

*Page 4, Confusing use of "one model" (line 18) and models in lines 19 - 23.*

Re-phrased (now page 4, line 22) to: "The aim of this paper is to explore the robustness of the
changes in transport found in the single-model studies described above, across an ensemble of
CCMs…"

Referee #3

*Overview.*

*The authors compare present-day day air pollution transport patterns with projected end-of-21st-
century conditions. For this purpose, seasonally-averaged volume mixing ratio fields of an artificial
tracer simulated by four chemistry-climate-model of the ACCMIP project are inter-compared. The
tracer has a 50-day lifetime and is emitted using present-day anthropogenic and biomass burning CO
emissions. The authors find a general decrease in tracer concentrations in the troposphere and an
increase near the tropopause in the simulation of all models. The authors attribute this mainly to
reduced convection in the tropics and an increase in tropopause height. The results agree across the
four models and also with previous studies.*

*General comments*

*The work is a scientifically sound study of the impact of climate change. Decrease in convective
activity, increase in tropopause height as well as weakening but increase in extend of the Hadley
circulation is a climate response simulated by many CCMs. Showing the impact on concentration
patterns as presented in the study is a step further in understanding the impact of these circulation
changes.*

We thank the reviewer for their positive comments.

*The relevance and novelty aspect of the study could be enhanced by elaborating in more detail the
agreement but also the disagreement with the studies from the literature, many of them mentioned
in the introduction. Please include this in the conclusion section.*

This is a good point. We have carefully gone through the manuscript and clarified further the
agreement and disagreement with previous studies. We have highlighted a further novelty of our
study being on seasonal transport patterns and their robustness across multiple models.

We have added more detail on agreement by providing quantitative estimates of decreases in CO-
tracer mixing ratios in the free troposphere to compare with previous studies and discussing
decreased convective mass fluxes in our study alongside findings by Held and Soden (2006) and
Abalos et al. (2017). We have emphasised in the discussion and conclusions the higher tropopause

height, and the poleward shift in zonal-mean winds and transport that have been outlined in sections 3.2 and 4.2.

The following additions have been made to the text:

*a) in terms of highlighting novelty:*

Page 4, line 22: "The aim of this paper is to explore the robustness of the changes in transport found in the single-model study described above, across an ensemble of CCMs participating in the recent Atmospheric Chemistry and Climate Model Intercomparison Project (ACCMIP) using a globally emitted CO-tracer (Lamarque et al., 2013), and to quantify for the first time seasonal transport changes in response to climate change and their dynamical attribution."

Page 11, line 27: This study quantifies the seasonal variation and the robustness of changes in transport under climate change. "

*b) in terms of highlighting agreement and disagreement:*

Page 3, line 31: "This study also showed…and a 25% lower tropospheric average tracer mixing ratio." (referring to Holzer and Boer 2001).

Page 8, line 1: "Fang et al. (2011) also find substantial decreases in annual-mean CO-tracer concentrations in the free troposphere (-2 to -12% at 400 hPa) but…"

Page 12, Discussion and conclusions, line 4: "The relative changes in annual-mean CO-tracer mixing ratios at the surface and in the free troposphere are of similar magnitude to those reported by Fang et al. (2011) using the GFDL-AM3 model. Somewhat larger decreases in tropospheric-average idealised tracer mixing ratios of 25% were reported in 2100 by Holzer and Boer (20001) under a different climate change scenario…"

Page 9, section 3.2, line 29: "A robust feature across all of the models is an overall reduction in convection, as reported by Held and Soden (2006), in response to climate change …"

Page 12, Discussion and conclusions, line 13: "…, in agreement with tropical convective mass flux reductions diagnosed by Held and Soden (2006). However, in contrast to our findings, Abalos et al.(2017) suggests decreases in convection mass fluxes are limited to ~5km."

Page 12, Discussion and conclusions, line 22: "The higher tropopause is a robust finding across climate change studies (e.g. Kang et al. 2014; Vallis et al. 2015)."

Page 12, Discussion and conclusions, line 25: "A poleward and upward shift in zonal-mean winds is consistent across the four models and noted in previous studies (e.g., Orbe et al. 2015)."

*The presented study focuses a lot on the tropics. But changes in the transport to the Arctic is also an important aspect given the impact of black carbon on the radiative forcing. For example Orbe et al. (2015) and (2013) find enhanced pole-ward transport towards the Arctic.  It would be good if the results of the current study would report more on the change in the poleward transport.*

We have considered the role of poleward transport in more detail, although we find in section 4.2 that much of the increase in CO-tracer mixing rations is due to the upward movement of the tropopause. We have plotted the difference in the vertical integrated CO-tracer column between 2000 and 2100 and find that this quantity increases north of ~30-40°N (as shown below). This shows that the re-distribution of CO-tracer in the future climate is not purely a vertical redistribution (else the net column change would be zero), and that some advective poleward transport appear to

occur. However, we have not added this extra figure to the text as we note above that the rise in the tropopause is the main driver of the CO-tracer increases in this region, although it is not possible to separate out the effects of these two processes cleanly.

[Figure]

*Figure: Differences in vertically integrated CO–tracer column mixing ratios between the 2090s-2000s for winter and summer.*

We have added further text of finding from Orbe et al. (2013, 2015) to the introduction and main text in section 3.2. :

Page 3, introduction, line 32: "More recently, Orbe et al. (2015) used idealized tracers of air-mass origin, as described in Orbe et al. (2013), to track how future increases in greenhouse…"

Page 8, section 3.2, line 7: "The contribution of the rise in the tropopause to the increase in CO-tracer mixing ratios is explored further in section 4.2. This near-tropopause increase in CO-tracer mixing ratios in the northern mid to high latitudes is also consistent with future increases in poleward transport reported by Orbe et al. (2015) based on tracers of air mass origin. Increases in the vertical integrated CO-tracer column between the 2000s and 2090s between 30°-40°N in all models also suggests an increase in advective transport poleward, since vertical re-distribution alone would not produce an increase in the vertical column."

*Also, please show maps from 90S -90N and do not omit the high latitudes.*

We have reproduced Figures 4-5 and 8-9 to display 90°S-90°N as requested at the end of these responses to the reviewers (see pages 14-17). However, there are no additional noteworthy CO-tracer or convective mass flux patterns at these high altitudes for this mid-tropospheric altitude range, and hence we prefer to retain the current latitude ranges of 40°S to 60°N for maximum clarity.  We do note that Figures 8, and 9 that show convective mass fluxes, originally had a more restricted latitude range so we have revised these figures so that the latitude ranges span 40°S to 60°N also.

*To avoid misunderstandings, the authors should discuss more clearly the limitations of the study because of the use of the artificial tracers: i.e. no impact of the photochemistry (loss by OH and production by VOC) or the precipitation patterns (deposition), and, most importantly, no change in the emissions.*

We have revised the text in the introduction to read more clearly:

Page 2, line 9: "To understand how these changes will influence future pollutant distributions, it is therefore important to disentangle the relative impacts of changes in transport and chemistry as well as future emission changes. The focus of this study is to quantify climate change impacts on atmospheric transport."

We have added the following text to the discussion and conclusions where we discuss our results in the context of air quality as requested by reviewer 1.

Page 13, line 17: "Hence, considering the impact of transport changes alone in the absence of stricter emissions controls, air quality in the future has the potential to be worsened in the vicinity of emission source regions especially in the tropics, due to reductions in vertical transport and dispersion by deep convection. However, this study examines the impacts of climate change on transport alone. Future air quality will also be greatly influenced by climate-driven changes in chemistry and by future changes in emissions."

*The end-of-century volume mixing ratios appear overall lower than the volume mixing ratios of the present day runs. But the total burdens should be the same ( p6 l1). It would be good if you could confirm the mass conservation and explain in more detail how the burden was redistributed. A comment on the mass conservation of the SL-advection scheme of STOC-HadAM3 might also be helpful.*

The global average tracer burdens are given below for the 2000s and 2100s periods in units of of kg-$CO/m^2$.

CMAM        *2000s*   0.000274696   *2100s*   0.000274829  difference= 0.05%

GISS         *2000s*   0.000273159   *2100s*   0.000273218  difference = 0.02%

UM-CAM      *2000s*  0.000270511   *2100s*   0.000270244  difference =  0.1%

STOC-HADAM3 *2000s*  0.000279364   *2100s*    0.000277364  difference = 0.7%

We believe these results re-affirm very-near mass conservation across the models, and do not feel that these differences merit further discussions for STOC-HadAM3.

We have however modified the text to read:

Page 6, line 13: "Note that the monthly-average atmospheric burden of the CO-tracer is virtually identical for the 2000s and 2090s, as expected…."

*The mechanism for the increase in tracer mixing ratios below the tropopause should be better explained. Is it simply because of the higher tropopause (i.e. the increase occurs near the present day tropopause) or do differences in stratosphere-troposphere exchange also play a role. Once it is established that the tropopause is higher from the GCM run, it is somewhat a trivial finding that tropospheric mixing ratio for primary tracers are increased at the same height.*

Our results in Figure 11 suggest that indeed it is simply the higher tropopause is the main cause increase in CO tracer mixing rations near and below the tropopause. We also suggest there may be a potential minor role for enhanced poleward transport based on an increase in the vertically integrated CO-tracer column poleward of ~ 30°N. However, Figure 11 shows this would not be more than a few ppb at most. We do not have any diagnostics to determine the role of STE, although Abalos et al. (2017) note this process also to be influenced by the displacement of the tropopause. However, we expect that with a 50-day lifetime there will be not be that much of the CO-tracer reaching the stratosphere, hence the amount of CO-tracer in the future might actually decrease with that mechanism if there is more downward transport of low CO values from the stratosphere.

We have added text to section 3.2:

Page 11, line 11: "Therefore, much of the CO-tracer increase near the tropopause that occurs in the future arises from a rise in tropopause height, as reported in Fang et al. (2011) and also by Abalos et al. (2017) using the e90 tracer….This also suggests that the impacts of enhanced poleward and upward transport in the northern mid-high latitudes near the tropopause on CO-tracer mixing ratios (section 3.2) are largely outweighed by the impact of the rise in tropopause; although these effects may be inter-related."

We have also revised the text in the Discussion and conclusions, to read:

Page 12, Line 23: "The strong increases in CO-tracer concentrations in the vicinity of the tropopause are mainly due to a higher tropopause under greenhouse gas warming, whereby this region has low-CO stratospheric air for present-day and higher-CO in tropospheric air in the future, in accord with Fang et al. (2011). A poleward and upward shift in zonal-mean winds is consistent across the four models and noted in previous studies (e.g., Orbe et al. 2015). Resultant enhanced poleward transport may also contribute in a minor fashion to CO-tracer increases in the future near the tropopause in the northern mid-latitudes; and changes in eddy mixing may also have an impact. However, all these processes may be inter-related such that it is not possible to discern the impacts of individual processes on CO-tracer mixing ratios. "

Page 12, line 32: "Further diagnostics to allow more detailed dynamical insights would be most useful to probe the relative contributions of different large-scale dynamical processes including stratosphere-troposphere exchange, alongside other aspects of the Hadley Circulation…"

*Specific comments:*

*P 1 L 24 , I think the weaker Hadley cell is a result of the GCM calculations, i.e. a given for this study So please consider rephrasing " .. in turn reflect..." to "...causes..."*

We have not been able to diagnose the Hadley circulation directly as we do not have the v wind component for calculation of the streamfunction, hence we are unable to make this definitive statement.

*P1 L25 Please add a sentence on the mechanism of increase in tracer because of increase in tropopause height.*

*The following text has been added to the abstract:*

*Page 1, line 26:* "…rise in tropopause height enabling lofting to higher altitudes"

*P2 L19-L27 This is more relevant for ozone and not so much for transport of primary pollutants. Consider shortening or omitting it.*

Agreed. The paragraph has been shortened (and one reference removed) to:

Page 2, line 19: "Descent from the lower stratosphere and upper troposphere to the mid-troposphere can occur in the dry intrusion airstreams of cyclones (e.g. Langford et al., 2015; Knowland et al. 2015). This is also the main mechanism for stratosphere–troposphere exchange of ozone that occurs in the mid-latitudes, and which may extend to the surface in regions prone to deep stratospheric ozone intrusions (Lin et al., 2015). Deep convection is also important for lofting surface pollution in mid-latitude regions in summer when the landmass is warm."

*P3 L 17 Is this response in ozone caused by transport ? If not please omit.*

This sentence relates shifts in storm track position to reduced mid- latitude cyclone frequency to reduced ozone, so yes this ozone response is caused by changes in transport. We have added "ozone" to the start of this sentence for clarity:

Page 3, line 13: "In terms of ozone pollution transport…"

*P3 L 23 Please discuss the impact of changes in stratosphere-troposphere exchange for the tracer transport*

*This text has been revised/added:*

   Page 3, line 19: "Ozone transport from the lower stratosphere to the troposphere will also be influenced by future changes in stratosphere-troposphere exchange, which is expected to increase under greenhouse gas warming owing to a strengthening of the Brewer Dobson circulation in the stratosphere, leading to higher ozone mixing ratios in the mid-to upper troposphere (Butchart and Scaife, 2001; Neu et al. 2014). Higher concentrations of tracers of stratospheric origin in the tropical/sub-tropical troposphere have been found due to enhanced stratosphere-troposphere exchange in a future warmer climate (Abalos et al. 2017). However, the effect of enhanced stratosphere-troposphere exchange on the concentrations of primary pollutants or tracers with no stratospheric source may be less important."

*P3 L31 Please add also Orbe et al. (2013).*

Now page 3, line 32: Added.

*P4 L29 Please add also information about the different convection schemes of the 4 Models*

The following text has been added and references included:

Page 5, line 3: "Deep convection schemes used by the models are based on two main parameterisations: Gregory and Rowntree (1990) for GISS-E2-R, UM-CAM and STOC-HadAM3 and Zhang and McFarlane (1995) for CMAM. In addition STOC-HadAM3 uses Collins et al. (2002) to derive using convective mass fluxes the probability of a parcel being subject to convective transport. Although these two parameterisations are based on a mass flux approach, there can be a wide spread in simulated convective mass fluxes within a single parameterisation (Scinocca and McFarlane 2004; Lamarque et al. 2013). In addition, how the transport of the CO-tracer is implemented will influence the impacts of the convection schemes."

*P5 L19 Please state the temporal resolution of the GFED 2 data*

Now page 5, line 29: Text amended to: "and monthly average biomass burning emissions". We have also updated the GFED v2 web-link in the paper.

*P7 L4 Please add a discussion here how the present-day CO tracers compare to actual CO. The NH CO maximum occurs in April, which seems not the case for tracer.*

*A* description of actual CO occurs just below this text on lines P7, L19-22. Here we have added the following text to discuss the season cycle of the CO-tracer and actual CO:

Page 7, line 22: "The seasonality of the CO-tracer and CO are fairly similar, with a more pronounced winter peak in the tropics in the CO-tracer in the mid-troposphere. The relative changes in CO-tracer mixing ratios are largest in the tropics and during winter and smaller in summer. Henceforth, the focus is on findings for boreal winter and summer."

*P7 L5 It is not clear that lower values do not deserve consideration. The changes could be even stronger. Please elaborate on this. MAM is the maximum of present day CO.*

"We thank the reviewer for noting this point. We erroneously stated that the absolute concentrations of the CO-tracer were lower during spring and autumn. We have revised the text accordingly describing the seasonality in absolute and relative (difference between 2090s-2000s) of the CO-tracer (as shown below). We also add text to outline that the peak differences the CO-tracer occurs in winter in the tropics and smaller differences occur in summer, which was our rationale for focussing on these two seasons. We initially had shown all four seasons but found the figures too repetitive:

[Figure]

*Figure: Annual cycle in CO–tracer mixing ratios at 500 hPa for the 2000s (top) and the differences 2090s-2000s (bottom).*

Page 6, line 19: "In the tropics, the largest CO-tracer mixing ratios occur during boreal winter (DJF), hereinafter winter, (Fig. 2) compared to boreal summer (JJA), hereinafter summer (Fig. 3). In the northern mid-latitudes CO-tracer mixing ratios are largest in spring and in the southern mid-latitudes in autumn. Elsewhere CO-tracer mixing ratios have a fairly uniform seasonal cycle."

Page 7, line 22: "The seasonality of the CO-tracer and CO are fairly similar, with a more pronounced winter peak in the tropics in the CO-tracer in the mid-troposphere. The relative changes in CO-tracer mixing ratios are largest in the tropics and during winter and smaller in summer. Henceforth, the focus is on findings for boreal winter and summer."

*P7 L23 Please clarify if this is the thermal present-day tropopause or the tropopause for the respective time slice. How does the thermal tropopause relate to a "tracer" tropopause?*

We have clarified in the methods section that a thermal tropopause is calculated for present-day and future and added a clarification to section 3.2 as requested that we are discussing the present-day tropopause.

Page 6 line 5: "The tropopause is defined separately as an average for the 2000s and the 2090s as the lowest model level at which the lapse-rate decreases to…"

Page 8, line 6: "Future CO-tracer mixing ratios also increase substantially by ~2-6 ppb (~10-25%) near the present-day tropopause and into…"

We have added text in the methods to discuss that lapse rate tropopause has been shown to compare well to the e90 tracer tropopause, and added appropriate references.

Page 6, line 6: "Studies have shown the lapse rate or thermal tropopause approximately coincides with a 90-day e-folding tracer tropopause which is used to distinguish stratospheric and tropospheric air (Prather et al. 2011; Abalos et al. 2017)."

*P8 L9 Please discuss also the changes at high latitudes. (see Orbe et al. 2013)*

As discussed in the general comments above, we have added further text on mid to high latitude changes to section 3.2:

Page 8, line 7: "This near-tropopause increase in CO tracer mixing ratios in the northern mid to high latitudes is also consistent with future increases in poleward transport, as reported by Orbe et al. (2015) based on tracers of air mass origin. Increases in the vertical integrated CO column between the 2000s and 2090s between 30°-40°N in all models also suggests an increase in advective transport poleward, since vertical re-distribution alone in this region would not produce an increase in the vertical column."

*P8 L17 From all ACCMIP models or only the four discussed here ?*

Text revised (now page 9, line 8) to: "from the four ACCMIP model simulations".

*P8 L18 For the present-day or end-of-century runs ?*

Text revised (now page 9, line 7) to: "For present-day, during both winter and summer…"

*P8 L20 Please provide more explanation for the up to factor 4 differences in the convective fluxes by the models. Is it driven by the meteorological input (i.e. T profile) or the specifics of the parametrisation.*

It is difficult to provide a conclusive explanation for these differences. Even if the large-scale temperature profiles were the same across the models there may still be differences in the behaviour of convective mass fluxes across the models. In response to an earlier comment for details on the convection schemes we added text from Scinocca and McFarlene (2004) that shows that even using a single convective parameterisation there can be a wide spread in behaviour due to implementation details. In previous work we have also noted large differences in convective mass fluxes simulated by different models (e.g., see fig 1. Doherty et al. 2005, Atmos. Chem. Phys.) We have added further text (and included references) to the methods and section 4 to discuss these points:

Page 5, line 3: "Deep convection schemes used by the models are based on two main parametrisations: Gregory and Rowntree (1990) for GISS-E2-R, UM-CAM and STOC-HadAM3 and Zhang and McFarlane (1995) for CMAM. In addition STOC-HADAM3 uses Collins et al. (2002) to derive using convective mass fluxes the probability of a parcel being subject to convective transport. Although these two parameterisations are based on a mass flux approach, there can be a wide spread in simulated convective mass fluxes within a single parameterisation (Scinocca and McFarlane 2004; Lamarque et al. 2013). In addition, how the transport of the CO-tracer is implemented will influence the impacts of the convection schemes."

(section 4, now page 9, line 15): "Substantial differences of a factor of 2-3 in annual-mean zonal convective mass fluxes simulated across three models (including STOC-HadAM3) were also reported in Doherty et al. (2005). Since, the same parametrization is used by UM-CAM, HadAM3 and GISS-ER-2, it may be the specific details of its implementation and interactions with internal parameters (Scinocca and McFarlane 2004) that cause this large difference in magnitudes across the four CCMs."

*P8 L21 Is there any indication, which of the models simulates more realistic mass fluxes.*

The distribution and strength of convection are not well constrained by observations so we cannot say which models are more realistic. All models will evaluate well for global precipitation patterns, but yet we see large differences in 3-D convective mass fluxes. In Doherty et al. (2005) we compared convective mass fluxes from HadAM3 to ERA-40, and found HadAM3 has higher mass fluxes (by up to a factor of 2) and that convection also generally reaches greater altitudes but it is not obvious that ERA-40 values reflect the real atmosphere either. It should be noted that even with this factor of four difference in convective mass fluxes, present-day CO tracer distributions and responses to climate change are fairly consistent.

*P9 L9 Please discuss also the increase in convection north of 60 N in DJF shown in Figure 6.*

This is already discussed at P9 L17 now Page 10, line 19. We have inserted "hemisphere" for clarity: "A strong increase in convective mass fluxes in the northern hemisphere polar latitudes …in winter"

*P10 L10 Please provide a plot of the increase in the thermal tropopause height between present-day and end-of-century or give some numbers in pressure and height.*

The sentence describing the rise in tropopause height refers to figs 2 and 3 that depict this graphically. In addition we have added the following text

now Page 11, line 3: " The annual-mean multi-model mean tropopause in the 2090s moves upward by 12.5 hPa in the tropics and 27.5 hPa in the mid-latitudes relative to is position in the 2000s."

*P10 L15 Please discuss Fig 11 in more detail. Is the conclusion based on the fact that dotted and solid blue lines overlap more than the respective green lines? I am not sure if this is actually the case especially for the tropics.*

Yes, that is correct. We have expanded and revised this text for clarity.

now page 11, line 9: "Comparison of annual-mean CO-tracer profiles reveals that when vertical CO-tracer profiles are compared in tropopause relative co-ordinates there is less difference between present-day and future, unlike when the CO-tracer profiles are plotted relative to pressure. Therefore, much of the CO-tracer increase near the tropopause that occurs in the future reflects a rise in tropopause height, as reported in Fang et al. (2011). This is evident for all models, in particular over the northern mid-latitudes (40°N) near the tropopause."

We have also revised the Figure 11 caption:

"plotted against altitude in pressure (green) and with distance from the tropopause for the respective time period (blue)."

*P10 L21 The weaker across-the-tropopause gradient is an interesting finding. It should be mentioned more clearly in the paper, i.e. the conclusions*

The weaker gradient response we discuss is within the mid-upper troposphere and is associated with the changes in zonal-mean winds. We have added the following text to the conclusions:

Page 12, line 26: "Resultant enhanced poleward transport may also contribute in a minor fashion to CO-tracer increases in the future near the tropopause in the northern mid-latitudes; and changes in eddy mixing may also have an impact. However, all these processes may be inter-related such that it is not possible to discern the impacts of individual processes on CO-tracer mixing ratios. "

*P11 L18 Please discuss also the response in the Arctic (compare with Orbe et al. (2013, 2015)) and the hemispheric gradient (Holzerand Boer, 2011)*

We have expanded this text to read:

Page 12, line 26: "A poleward and upward shift in zonal-mean winds is consistent across the four models and noted in previous studies (e.g., Orbe et al. 2015). Resultant enhanced poleward transport may also contribute…" (see comment above).

In response to reviewer 2 general comments we have added text or reduced tropospheric average mixing ratios from Holzer and Boer (2001):

Page 12, line 6: "Somewhat larger decreases in tropospheric-average idealised tracer mixing ratios of 25% were reported in 2100 by Holzer and Boer (20001) under a different climate change scenario and attributed to a higher tropopause."

*P11 L22 Please mention that you found a weaker across-the-tropopause gradient (if this is the case)*

As noted above, we did not discuss a cross-tropopause gradient, only a weaker gradient in the upper troposphere in Fig 11, so we retain only to mention a change in mixing:

Page 12, line  28 "…and changes in eddy mixing may also have an impact."

*Figures 4,5, 8 etc. please show maps from 90S-90N*

Please see our response in the general comments. We have reproduced these Figures (see pages 14-17 of these responses). However, there are no additional noteworthy CO-tracer or convective mass flux patterns at these high altitudes for this mid-tropospheric altitude range, and hence we prefer to retain the current latitude ranges of 40°S to 60°N for maximum clarity.

*Figure 1 add to caption "for different seasons"*

Added.

*Figure 2, please use either present-day or REF not both*

Amended to present-day.

*Figure 11, please mention that the distance is from the respective tropopause for each time slice.*

Added.

Literature:  Orbe, C., M. Holzer, L. M. Polvani, and D. Waugh (2013), Air-mass origin

as a diagnostic of tropospheric transport, J. Geophys.  Res.  Atmos.  118, 1459–1470,

doi:10.1002/jgrd.50133

Added.

*Figure 4 . Top panels within subplots a-d): Present-day (1995-2006) DJF climatological mean CO distribution, averaged over 400-800 hPa.  Bottom panels: 2090-2099 (RCP8.5) - 1995-2006 difference in DJF climatological mean distributions, wherein black contours denote the present-day climatology.  Results are presented for a) UM-CAM and b) STOC-HadAM3 (top panels) and c) CMAM and d) GISS-E2-R (bottom panels). Grey shading indicate where results are not significant at p < 0.05 as evaluated with a Student t-test using 10 years of data for the 2090s  (RCP 8.5) and present-day climate simulations. Note the different scales for UM-CAM and STOC-HadAM3 and for CMAM and GISS-E2-R for the difference plots.*

[Figure]

*Figure 5: Same as Figure 4 but for JJA. Note the different scales for UM-CAM and STOC-HadAM3 and for CMAM and GISS-E2-R for the difference plots.*

[Figure]

*Figure 8. Top panels within subplots a)-d): DJF climatological mean convective mass fluxes, averaged over 300-800 hPa for 1996-2005 (present-day).  Bottom panels e)-h): Same, but for 2090-2099 (RCP8.5)- 1996-2005 (present-day) difference, wherein black contours denote the present-day climatology.  Results are presented for UM-CAM and STOC-HadAM3 (top panels) and CMAM and GISS-E2-R (bottom panels). Grey shading indicate where results are not significant at p < 0.05 as evaluated with a Student t-test using 10 years of data for the 2090s (RCP 8.5) and present-day climate simulations.*

[Figure]

*Figure 9. Same as Figure 8 but for JJA. Note the different scales for UM-CAM and STOC-HadAM3 and for CMAM and GISS-E2-R for the difference plots.*

[Figure]